# Impact of Betaine Supplementation on Growth Performance, Nutrient Digestibility, Egg Quality, Gas Emissions, and Blood Profile in Laying Hens Under Heat Stress Conditions

**DOI:** 10.3390/ani15172565

**Published:** 2025-08-31

**Authors:** Zhenyu Ding, In Ho Kim

**Affiliations:** 1Department of Animal Biotechnology, Dankook University, Cheonan 31116, Choongnam, Republic of Korea; d72231721@dankook.ac.kr; 2Smart Animal Bio Institute, Dankook University, Anseodong, Cheonan 31116, Choongnam, Republic of Korea; 3School of Media, Tonghua Normal University, Yucai, Dongchang, Tonghua 134002, China

**Keywords:** betaine, blood profile, egg production, egg quality, gas emission, heat stress

## Abstract

During the hot summer season, high temperatures often persist, which increases the egg-laying burden on laying hens. To alleviate the burden of heat stress, we supplemented the feed of laying hens with betaine. Betaine boosts the egg production rate by aiding methyl metabolism, supplying methyl groups for amino acid synthesis and offsetting reduced intake due to heat. Supplementing the summer feed of laying hens with betaine is beneficial for increasing the egg production rate and egg quality; improving the emission of harmful gases; and reducing the level of plasma cortisol. Therefore, supplementing the feed of laying hens with betaine in summer can alleviate heat stress; improve the hens’ health status, production performance, physiological state and nutritional metabolism; and has a beneficial effect on laying hens in high-temperature environments.

## 1. Introduction

Summer heat stress poses a significant challenge to the poultry industry. In environments with high temperatures and humidity, the thermoregulatory ability of poultry is significantly reduced. As a result, the adverse impacts of heat stress on their productive performance are amplified [1]. This heat stress gives rise to a series of issues in laying hens [2], broilers [3], and ducks [4], such as the decline in production performances, deterioration of egg or meat quality, and increase in morbidity. Consequently, it inflicts substantial economic losses on the poultry-breeding industry.

Betaine, a by-product of sugar beet processing, shows great potential as a feed additive for alleviating heat stress in animals [5]. Betaine, which has osmoprotective properties; acts as an osmoregulator to maintain normal cellular osmotic pressure; prevent water loss; protect intestinal cell proteins and enzymes from stresses like high temperatures; regulate water balance; stabilize tissue metabolism; reduce performance loss; and alleviate heat stress physiological responses by improving nutrient digestibility [6]. In plants, Betainee is synthesized and accumulates as an osmoprotectant against salt and temperature stress. In animals, betaine is a product of choline oxidation and plays a crucial role in regulating osmotic pressure [7]. Several studies have shown that dietary betaine supplementation can help regulate osmotic pressure in animals exposed to high-temperature stress [8]. This regulation helps to prevent dehydration, retain cellular water [7], and improve both digestibility and growth performance [9]. Heat stress raised rectal temperature and reduced egg production, weight and quality in hens. Betaine lowered rectal temperature by 0.4 °C, which was an improvement, and enhanced feed conversion and survival, but not intestinal or reproductive indices [10]. Numerous studies have reported various beneficial effects of dietary betaine supplementation in poultry. For example, later in the laying period, it has been found to increase the laying rate and eggshell thickness [11]. In broiler chickens, it influences the digestibility of dry matter and nitrogen [4]. Additionally, it has been shown to increase average egg production and plasma cortisol levels [12]. In Japanese quail, dietary betaine supplementation impacts growth performance, carcass traits, and blood chemistry [13]. However, a comparison of two laying hen strains under acute and chronic heat stress, examining betaine-supplemented water and feed restriction, showed that white hens were more severely affected by heat stress than brown hens, with neither measure improving their heat stress indicators [14].

In response to the global call for carbon neutrality, reducing harmful gas emissions from the livestock industry is urgent [15]. A study by the FAO indicates that livestock contributes significantly to Greenhouse Gas emissions [16]. Immediate action is needed. In recent years, however, betaine has been found to reduce the emission of harmful gases from the feces of laying hens under heat stress. Based on the betaine-related findings, we hypothesized that betaine, functioning as a heat stress-mitigating agent which also reduces harmful gas emissions, would have beneficial effects on laying hens subjected to high-temperature conditions. Therefore, the objective of this study was to evaluate the impact of dietary betaine supplementation on the growth performance, nutrient digestibility, egg quality, harmful gas emissions, and blood profile of laying hens during the summer season.

## 2. Materials and Methods

### 2.1. Birds, Husbandry Management, and Diets

A total of 216 Hy-Line brown laying hens, all 28 weeks of age, were randomly assigned to 3 different diet treatment groups, with 6 replicates per group and 12 laying hens per replication. Hens were housed in single pens, each with a length of 50 cm, a width of 38 cm, and a height of 40 cm. This trial lasted for 8 weeks during the summer months of July and August. The lighting period was a continuous schedule of 20 lux with 16 hours of light (05:00 to 21:00) and 8 hours of darkness. The experiment environment temperature was controlled under cyclic heat stress at 32 ± 1 °C from 9:00 am to 5:00 pm, with a rest time at a temperature at 26 ± 1 °C and 60–70% relative humidity. The cyclic temperature was controlled using a computerized environmental control system (Model: SMC-100, Seoul Machinery Co., Ltd., Seoul, Republic of Korea) equipped with precision heaters, exhaust fans, and temperature/humidity sensors. Each pen was equipped with nipple drinkers and movable trough feeders, allowing the hens to access feed and water freely. The experimental basal diet with National Research Council standards for laying hens [17] and the composition of laying hen basal diet are presented in Table 1. Experimental diets include (1) CON: (basal diet); (2) TRT1: basal diet + betaine 0.075%; and (3) TRT2: basal diet + betaine 0.15%. betaine (Betafin^®^, Betaine anhydrous, 97%).

### 2.2. Sampling and Analysis

The experiment included the recording of daily feed intake (FI) and the total number of eggs laid and the percentage of downgraded eggs (those with defects or abnormalities) for each replication. The egg production percentage and feed conversion ratio (FCR) were calculated daily based on the recorded data of each copied record. Egg quality evaluation was conducted at the end of the 4th and 8th weeks during the experimental period. Egg quality was checked weekly from 1 to 8 weeks. At 17:00 h, 18 eggs (3 eggs per replication, excluding damaged eggs) were selected from each treatment; there were no shell defects or cracks. Egg quality measurements were carried out on the same day at 20:00 h. Experts used an eggshell color fan (DSM, Basel, Switzerland) to provide scores for the color and visual appearance of the eggshells. The strength of the eggshells was evaluated by an eggshell force tester II (Robotmation Co., Ltd., Tokyo, Japan), and the thickness of the eggshells was measured by a dial-type tubular thickness gauge (Ozaki MFG. Co., Ltd., Tokyo, Japan). The calculation method focused on the average thickness of the round end, pointed end, and middle part of the egg, excluding the inner membrane. Haugh Units (HU), albumen height, and yolk color were determined using an egg multi-tester (Touhoku Rhythm. Co., Ltd., Fukushima, Japan).

### 2.3. Sampling and Analysis of Nutrient Digestibility

Seven days before the experiment’s end, 0.3% of chromic oxide was added to the experiment’s feed as an indigestible marker to determine the apparent nutrient digestibility (AND), dry matter (DM), nitrogen (N), and energy (E). Following the incorporation of chromium (Cr_2_O_3_), triplicate feed samples were aseptically collected from each treatment group and placed in pre-sterilized plastic containers. Samples were stored under anaerobic conditions pending further biochemical analysis. During the final 72 h of the experimental period, total excreta outputs were aggregated per replicate pen, homogenized using a sterile spatula, and sub-sampled. Excreta aliquots were immediately flash-frozen in liquid nitrogen and stored at −20 °C for subsequent determination of nutrient digestibility coefficients DM, N, and E. Prior to the instrumental analysis, excreta specimens underwent dehydration in a forced-air convection oven at 105 °C for 32 h until they reached a constant weight. Dried samples were then pulverized using a cutting mill (Retsch ZM 200) to achieve a particle size of ≤1 mm. Feed and excreta proximate analyses were conducted in triplicate according to AOAC International methods (2003). Chromium concentration was quantified via UV-Vis spectrophotometry (Shimadzu UV-1201, Kyoto, Japan) at a wavelength of 372 nm. Gross energy was determined using an isoperibol oxygen bomb calorimeter (Parr 6100, Moline, IL, USA) standardized with benzoic acid. Total nitrogen content was measured via the Dumas combustion method using a LECO FP-528 nitrogen analyzer (LECO Corporation, St. Joseph, MI, USA). AND coefficients were calculated using the following equation:

AND (%) = 100 − [(nutrient concentration in excreta/nutrient concentration in diet) × (chromium concentration in diet/chromium concentration in excreta) × 100]

### 2.4. Sampling and Analysis of Gas Emission

After the experiment, excreta samples were carefully collected from each pen and thoroughly mixed. These samples were then placed into 2.6 L plastic boxes in duplicate. Each box was specifically designed with a small hole located precisely in the middle of one sidewall, which was initially sealed tightly with adhesive tape. Subsequently, the samples were left undisturbed to ferment for 5 days at a controlled temperature of 25 degrees Celsius. At the conclusion of the fermentation phase, gaseous emissions were quantified via a GV-100 gas sampling pump (Gastec Corp., Kanagawa, Japan) in conjunction with colorimetric detection tubes. Ammonia (NH_3_), hydrogen sulfide (H_2_S), methyl mercaptans, acetic acid, and carbon dioxide (CO_2_) were analyzed using detection tubes No. 3L, No. 4LT, and No. 70L (Gastec Corp., Kanagawa, Japan), respectively. All measurements were performed in triplicate to ensure analytical reliability. To achieve this objective, the seal was carefully pierced, and precisely 100 mL of headspace air was sampled at a height of approximately 2 cm above the excreta. Once the air-sampling process was completed, each box was promptly resealed and securely covered with adhesive tape. The headspace measurements were then meticulously carried out again after a time interval of 58 h. The gas content was accurately calculated as the average value of the two measurements.

### 2.5. Serum Analysis

At the final stage (week 8) of the trial, blood samples were collected for serum hormone analysis. A total of 36 hens were involved, with 12 hens per treatment group (2 hens per replicate). Venous blood samples were aseptically collected via brachial venipuncture into non-heparinized vacuum tubes (Becton Dickinson Vacutainer Systems, Franklin Lakes, NJ, USA). All blood draws were performed by trained phlebotomists following standard aseptic protocols to minimize the risk of contamination. To minimize the stress on the animals and ensure sample consistency, the blood collection from each hen was completed within a time frame of no more than 3 min from the moment of capture. To mitigate the potential confounding effects of circadian rhythm on hormonal profiles, all blood collections were standardized to occur between 11:00 and 12:00 h across experimental days. Following venipuncture, samples were immediately centrifuged at 3000× *g* for 15 min at 4 °C to isolate serum fractions. Separated sera were then aliquoted into sterile microcentrifuge tubes and stored at −20 °C until assayed. Serum cortisol concentrations were quantified using a solid-phase radioimmunoassay (RIA) kit (Coat-A-Count TKCO, Diagnostic Products Corporation, Los Angeles, CA, USA) according to the manufacturer’s standardized protocol. All assays were performed in duplicate, and intra-assay coefficients of variation were maintained below 10% to ensure analytical precision.

### 2.6. Statistical Analysis

All data were analyzed using the General Linear Models (GLM) procedure within the SAS 9.4 software (SAS Institute Inc., Cary, NC, USA) in a randomized complete-block design framework. The replications, with *n* = 12 birds, were designated as experimental units. Orthogonal comparisons were conducted using polynomial regression to determine Linear and quadratic effects on the grade level of betaine supplement in laying hens’ diet. Variability in the data was expressed as the standard errors mean. Differences among treatment means were determined using Tukey’s range test for overall *p*-value. The *p* < 0.05 was considered as significant and *p* < 0.10 was considered as a trend.

## 3. Results

### 3.1. Growth Performance

The effects of betaine on the growth performance of laying hens are presented in Table 2. The results demonstrated that the inclusion of betaine in the laying hens’ diet significantly linearly increased egg production during week 6, week 7 and week 8. And FCR showed a significant tendency (*p* < 0.10) at week 8.

### 3.2. Egg Quality

The effects of betaine on the egg quality of laying hens are presented in Table 3. The HU height exhibited a linear improvement in week 8 with the supplementation of betaine.

### 3.3. Nutrient Digestibility

The effects of betaine on the nutrient digestibility of laying hens are presented in Table 4. During week 8 of the experiment, there was a significant increase in the digestibility of DM, N.

### 3.4. Gas Emission

The effects of betaine on the harmful gas emission of laying hens are presented in Table 5. As shown in Table 5, regarding the effects of betaine on harmful gas emissions from laying hens, there was a linear reduction in the emissions of NH_3_, H_2_S, and methyl mercaptans in week 8 when the diet was supplemented with betaine.

### 3.5. Blood Profile

The effects of betaine on the cortisol of laying hens are presented in Table 6. Laying hens fed a diet supplemented with betaine showed linear decreased cortisol levels in the blood and quadratic cortisol levels.

## 4. Discussion

The present study was designed to determine the effects of betaine on the growth performance, nutrient digestibility, egg quality, harmful gas emissions, and blood profile of laying hens during the summer season. Betaine acts as an osmoprotectant; animals absorb it through the duodenum of the small intestine, where it accumulates in cells without disrupting cellular function, thereby protecting them against osmotic stress [18]. On the other hand, betaine serves as an important methyl donor in feed, acting as a catabolic source of methyl groups through transmethylation. The transmethylation of betaine occurs in the mitochondria of the liver and kidneys via the methionine cycle [19]. This process participates in the remethylation of homocysteine, which helps eliminate toxic metabolites and preserve methionine [20], thus contributing to the maintenance of the body’s metabolic needs. Methionine, metabolized in the liver, is essential for protein synthesis and the regulation of cell function. In recent years, the application of betaine has been reported to enhance poultry’s resistance to heat stress. In a study involving 32-week-old laying hens, supplementation of 3.0 or 6.0 g/kg of betaine to the basal diet under high-temperature conditions resulted in an upward trend in egg production and a significant improvement in eggshell strength [21]. Under natural summer temperatures during July and August, when laying hens’ diet was supplemented with 0.7 g/kg or 1.5 g/kg of betaine, both supplemental levels of betaine increased the average egg production [12]. Laying hens in the later stages of production were given dietary betaine in proportions of 0.1% or 0.5%, and those which received 0.5% betaine laid more eggs with thicker eggshells [11]. Dietary supplementation with 1 g/kg or 3 g/kg of betaine improved laying performance in aged laying hens [22]. The broilers whose diets were supplemented with 0.5 g/kg, 1 g/kg, and 2 g/kg betaine under long-term heat stress showed improved BWG and feed intake [3]. Supplementing the diets of quails with 0.75 g/kg, 1.5 g/kg, or 2.25 g/kg or betaine had a more significant effect on weight gain and improved FCR [13]. The effects of betaine described in the papers above are the same as those in this experiment. Meanwhile, it has been shown that supplementation with 0.15% betaine can relieve heat stress and improve egg production by 1.5–1.9% (weeks 6–8) under hot summer condition, directly enhancing economic benefits.

Egg quality is a vital consideration in the poultry sector, and high summer temperatures influence the quality of laying hens. Kim, H.-R., C. Ryu [1] reported that severe heat stress affected the egg yolk color, eggshell thickness and strength, and HU of laying hens’ eggs. As [23] presented, supplementing laying hens’ diets with betaine (0.5 g/kg) increased egg weights and improved yolk color and albumin width at 57 week. Supplementing laying hens’ feed with 1 g/kg of betaine significantly increased the shell thickness, HU, and egg weight [24]. HU is a measure of the albumen quality index: the higher the value, the fresher the egg and the higher its quality [25]. In our study, betaine supplementation had a significant impact on HU at week 8. The diverse outcomes may be attributed to several factors, including varying dietary compositions, types of betaines, variations in dosage, the duration of the study, heat stress temperatures, and genetic variations among hens. One limitation of this study is that only two folate concentrations (0.075% and 0.15%) were used, which prevented us from determining the exact optimal dose. This might have led to only HU appearing to have significant effects on egg quality. Future studies will set up three to four doses (such as 0.05%, 0.1%, 0.15%, 0.2%) to establish a complete dose–response curve. Additionally, comparing betaine with other heat stress alleviators (such as vitamin C and probiotics) might provide more options for the industry.

Betaine potentially affects nutrients’ digestibility, and it may be that the permeability of betaine plays a role in supporting intestinal cell growth and survival and enhancing cell activity [7]. Supplementation of betaine in the diet of laying hens under a high-temperature environment selectively changed the expression of the jejunum tight junction-related gene [21]. In the late laying period, the villi of jejunum and ileum supplemented with betaine were well organized, and the villus height and villus height/crypt depth ratio of jejunum and ileum were significantly increased [22]. The organic matter crude protein ether extract crude fiber and nitrogen-free extract digestion significantly increased as the quantity of betaine in the broiler diets increased [26]. Betaine supplementation acts as a “methyl donor” in broiler diets to preserve the energy needed for Na+/K+ pumps at high temperatures, and may promote the growth of broilers by consuming this energy [27,28]. In a recent report, adding betaine to broilers’ diets increased the digestibility of DM and N [3,29]. Growing pigs under heat stress also exhibited significant digestibility in DM and N [30]. These results were similar to this experiment’s results, indicating that betaine has been shown to have a positive effect on nutrient digestibility under heat stress conditions. However, Al-Qaisi [10] reported that betaine improved egg production but not nutrient digestibility. In contrast, our study found that betaine enhanced DM and N digestibility, which we attribute to the cyclic (vs. constant) heat stress model—cyclic stress may be less severe, allowing betaine to exert its osmoprotective effect on intestinal cells [7] and improve nutrient absorption. In this study, harmful gas emission results indicated that betaine supplementation reduced the levels of ammonia (NH_3_), hydrogen sulfide (H_2_S), and methyl mercaptans in the excreta of laying hens exposed to heat stress. The improvement in the digestibility of dry matter (DM) and nitrogen (N) suggests that betaine may enhance nutrient utilization, thereby contributing to the reduction in these harmful gas emissions. This finding was similar to the results of a study on broilers [31], but we found no studies which addressed the effect of betaine on the harmful gas emissions of laying hens under heat stress, so we speculate that betaine supplementation can improve barn air quality, reducing respiratory diseases in hens and workers.

When poultry is subjected to heat stress, its thermoregulatory system detects temperature changes and relays this information to the hypothalamus [32]. Subsequently, the hypothalamus secretes corticotrophin-releasing hormone (CRH). This CRH acts on the pituitary gland, prompting it to secrete adrenocorticotrophin (ACTH). In turn, ACTH stimulates the adrenal cortex to secrete cortisol. Through this process, heat stress activates the hypothalamic–pituitary–adrenal (HPA) axis, thereby resulting in increased cortisol secretion [33]. If the duration or intensity of heat stress is excessive, the dehydration and electrolyte imbalances resulting from heat stress can impact the function of the HPA axis [34,35]. Prolonged exposure to high levels of cortisol can exert detrimental effects on the body [36]. It reduces the body’s tolerance to heat stress and heightens the risk of heat stress-related diseases, such as changes in the lymphocyte ratio, immune functions, digestibility, glucose homeostasis, and nutrient metabolism [37,38,39,40,41]. Heat stress leads to a decrease in the number of lymphocytes (L) and an increase in the number of heterophile granulocytes (H), increasing in the H:L ratio [42]. Betaine decreases the H number and increases the L number, stimulating the release of hypothalamic corticotrophin-releasing hormone [43]. Cortisol, an important component of adrenocortical hormones, serves as a biomarker of stress. In the heat stress context, the measurement of cortisol in poultry is a primary means of assessing hypothalamic corticotropin levels [1]. Some reports have shown that betaine supplementation can mitigate the adverse effects related to cortisol. Under natural summer temperatures, betaine supplementation of laying hens reduced cortisol levels [12]. Gudev, D., S. Popova-Ralcheva [44] reported that supplementing betaine with air ammonia concentration in a broiler diet decreased plasma cortisol levels. Supplementation of betaine also exhibited a moderating effect on plasma cortisol levels in growing–finishing pigs under heat stress [30]. In contrast, De Baets [14] found that betaine had no effects on cortisol in white hens, while our study (using brown hens) showed a significant cortisol reduction. This discrepancy may be due to strain differences—brown hens (Hy-Line brown) are more heat-tolerant than white hens, so betaine’s stress-alleviating effects are more pronounced in brown hens, which is valuable information for farms raising brown laying hens. Furthermore, the quadratic trend of cortisol in this study suggests that 0.075–0.15% betaine balances cost-effectiveness and stress alleviation. Thus, betaine is a cost-efficient, eco-friendly feed additive for heat-stressed laying hens.

## 5. Conclusions

Betaine supplemented in the diet of laying hens has been advantageous for improving egg production and egg quality and reducing harmful gas emissions and the negative effects of plasma cortisol levels in summer. Therefore, betaine can be used in laying hens during summer to alleviate heat stress and improve health, productive performance, physiological conditions, and nutrient metabolism.

## Figures and Tables

**Table 1 animals-15-02565-t001:** Composition of laying hen diets (as-fed basis).

Item	Basal Diet
Ingredients (%)	
Corn	61.39
Soybean meal	16.39
Corn gluten meal	4.94
Wheat bran	3.60
Tallow	0.97
MDCP	1.65
Limestone	9.75
Salt	0.25
Lysine (80%)	0.30
Methionine (50%)	0.26
Vitamin mix ^1^	0.20
Mineral mix ^2^	0.20
Choline (50%)	0.10
Total	100.00
Calculated value	
ME, kcal/kg	2800
Crude Protein, %	16.50
Crude Fat, %	3.43
Crude Fiber, %	2.89
Crude Ash, %	4.57
Calcium, %	4.08
Phosphorus, %	0.68
Available Phosphorus, %	0.45
Lysine, %	0.88
Methionine, %	0.42
Methionine + Cystine, %	0.78

^1^ Provided per kg of diet: vitamin A—8000 IU; vitamin D3—3300 IU; vitamin E—20 g; vitamin K3—2.5 g; vitamin B1—2.5 g; vitamin B2—5.5 g; vitamin B6—4 g; vitamin B12—23 mg; biotin—75 mg; folic acid—0.9 g; niacin—30 g; D-calcium pantothenate—8 g. ^2^ Provided per kg of diet: Fe—40 g as ferrous sulfate; Cu—8 g as copper sulfate; Mn—90 g as manganese oxide; Zn—80 g as zinc oxide; 1.2 g as potassium iodide; and Se—0.22 g as sodium selenite.

**Table 2 animals-15-02565-t002:** The effects of betaine on growth performance in laying hens ^1^.

Items	CON	TRT1	TRT2	SEM ^2^	Linear	Quadratic
Body weight, g						
Initial	1909	1910	1907	1	0.330	0.215
Week 4	1921	1922	1924	2	0.427	0.999
Week 8	1939	1940	1943	5	0.580	0.854
Week 1						
Downgraded egg, %	0.6	0.6	0.6	0.3	0.969	0.947
Egg production, %	95.4	95.4	95.6	0.6	0.826	0.899
FCR	1.87	1.87	1.88	0.01	0.837	0.940
Week 2						
Downgraded egg, %	0.7	0.6	0.6	0.3	0.943	0.997
Egg production, %	95.2	95.4	95.6	0.7	0.653	0.999
FCR	1.89	1.89	1.88	0.02	0.759	0.807
Week 3						
Downgraded egg, %	0.7	0.6	0.6	0.4	0.908	0.941
Egg production, %	94.8	95.2	95.4	0.8	0.550	0.908
FCR	1.91	1.90	1.89	0.01	0.401	0.948
Week 4						
Downgraded egg, %	0.8	0.7	0.6	0.4	0.703	0.918
Egg production, %	94.4	95.0	95.2	0.8	0.480	0.837
FCR	1.93	1.92	1.91	0.02	0.318	0.985
Week 5						
Downgraded egg, %	0.8	0.7	0.7	0.4	0.817	0.905
Egg production, %	94.0	94.6	95.0	0.8	0.355	0.913
FCR	1.95	1.93	1.92	0.01	0.173	0.640
Week 6						
Downgraded egg, %	0.9	0.8	0.8	0.3	0.909	0.966
Egg production, %	93.3 ^b^	94.0 ^ab^	94.6 ^a^	0.4	0.031	0.847
FCR	1.96	1.94	1.92	0.01	0.931	0.126
Week 7						
Downgraded egg, %	1.1	0.9	0.8	0.3	0.6802	0.915
Egg production, %	92.7 ^b^	94.0 ^a^	94.2 ^a^	0.4	0.011	0.321
FCR	1.97	1.94	1.93	0.02	0.111	0.511
Week 8						
Downgraded egg, %	1.2	1.0	0.8	0.3	0.439	0.924
Egg production, %	92.7 ^b^	93.9 ^ab^	94.2 ^a^	0.4	0.035	0.412
FCR	1.97	1.95	1.94	0.01	0.063	0.920

^1^ Abbreviation: CON—basal diet; TRT1—basal diet + betaine 0.075%; TRT2—basal diet + betaine 0.15%. ^2^ Standard error of means. ^a,b^ Means in the same row with different superscripts differ significantly.

**Table 3 animals-15-02565-t003:** The effects of betaine on egg quality in laying hens ^1^.

Items	CON	TRT1	TRT2	SEM ^2^	Linear	Quadratic
Week 4						
Eggshell color	10.9	11.0	10.7	0.3	0.785	0.695
HU	89.9	90.4	91.0	1.5	0.582	0.969
Egg weight, g	62.1	62.2	62.4	1.0	0.852	0.979
Yolk color	7.2	7.4	7.5	0.3	0.377	0.889
Albumen height, mm	9.5	9.9	10.4	0.4	0.182	0.942
Eggshell Strength, kg/cm^2^	4.06	4.09	4.13	0.20	0.803	0.999
Eggshell Thickness, mm^2^	38.1	38.2	38.5	0.7	0.625	0.935
Week 8						
Eggshell color	10.8	10.5	11.0	0.4	0.695	0.457
HU	86.0 ^b^	88.2 ^ab^	89.9 ^a^	1.3	0.025	0.850
Egg weight, g	62.7	62.8	63.3	0.8	0.545	0.836
Yolk color	7.5	7.6	7.8	0.2	0.267	0.830
Albumen height, mm	9.3	9.8	10.2	0.5	0.272	0.908
Eggshell Strength, kg/cm^2^	4.01	4.06	4.11	0.16	0.968	0.989
Eggshell Thickness, mm^−2^	37.8	38.0	38.3	0.7	0.608	0.984

^1^ Abbreviation: CON—basal diet; TRT1—basal diet + betaine 0.075%; TRT2—basal diet + betaine 0.15%. ^2^ Standard error of means. ^a,b^ Means in the same row with different superscripts differ significantly.

**Table 4 animals-15-02565-t004:** The effects of betaine on nutrient digestibility in laying hens ^1^.

Items	CON	TRT1	TRT2	SEM ^2^	Linear	Quadratic
Week 8						
Dry matter (%)	72.24 ^b^	74.08 ^ab^	75.87 ^a^	1.37	0.035	0.984
Nitrogen (%)	57.52 ^b^	59.4 ^ab^	60.04 ^a^	1.19	0.043	0.523
Energy (%)	72.43	74.22	75.17	1.45	0.087	0.749

^1^ Abbreviation: CON—basal diet; TRT1—basal diet + betaine 0.075%; TRT2—basal diet + betaine 0.15%. ^2^ Standard error of means. ^a,b^ Means in the same row with different superscripts differ significantly.

**Table 5 animals-15-02565-t005:** The effects of betaine on harmful gas emission in laying hens ^1^.

Items	CON	TRT1	TRT2	SEM ^2^	Linear	Quadratic
Week 8						
NH_3_ (ppm)	22.58 ^a^	22.08 ^a^	20.75 ^b^	0.44	0.009	0.443
H_2_S (ppm)	5.02 ^a^	4.53 ^ab^	3.90 ^b^	0.26	0.010	0.822
Methyl mercaptans (ppm)	11.92	11.50	9.58	0.73	0.050	0.441
Acetic acid (ppm)	9.67	9.75	9.50	0.82	0.907	0.893
CO_2_ (ppm)	1650	1667	1633	122	0.918	0.858

^1^ Abbreviation: CON—basal diet; TRT1—basal diet + betaine 0.075%; TRT2—basal diet + betaine 0.15%. ^2^ Standard error of means. ^a,b^ Means in the same row with different superscripts differ significantly.

**Table 6 animals-15-02565-t006:** The effects of betaine on blood profile in laying hens ^1^.

Items	CON	TRT1	TRT2	SEM ^2^	Linear	Quadratic
Week 8						
Cortisol, ng/ml	2.76 ^b^	2.24 ^a^	2.19 ^a^	0.11	0.0005	0.066

^1^ Abbreviation: CON—basal diet; TRT1—basal diet + betaine 0.075%; TRT2—basal diet + betaine 0.15%. ^2^ Standard error of means. ^a,b^ Means in the same row with different superscripts differ significantly.

## Data Availability

The data are available on request due to privacy.

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
