# Peer review of "Impact of Betaine Supplementation on Growth Performance, Nutrient Digestibility, Egg Quality, Gas Emissions, and Blood Profile in Laying Hens Under Heat Stress Conditions"

_animals, 2025, doi:10.3390/ani15172565_

Round 1

Reviewer 1 Report

Comments and Suggestions for Authors

The present study investigated the effects of betaine supplementation on growth performance, nutrient digestibility, egg quality, gas emissions, and blood profile in laying hens under heat stress. The results showed that betaine supplementation could enhance the egg production and egg quality during the summer season. Following are some comments on this manuscript.

Major: The major problem for this manuscript is the novelty. Several studies related with the effects of betaine supplementation on laying performance of hens under heat stress have been published [1-3]. What new findings have the authors found?

[1]        DE BAETS R, BUYSE K, ANTONISSEN G, et al. Betaine and feed restriction as potential mitigation strategies against heat stress in two strains of laying hens [J]. Poultry Science, 2024, 103(10): 104104.

[2]        AL-QAISI M, ABDELQADER A, ABUAJAMIEH M, et al. Impacts of dietary betaine on rectal temperature, laying performance, metabolism, intestinal morphology, and follicular development in heat-exposed laying hens [J]. Journal of Thermal Biology, 2023, 117: 103714.

[3]        RATRIYANTO A, MOSENTHIN R. Osmoregulatory function of betaine in alleviating heat stress in poultry [J]. Journal of animal physiology and animal nutrition, 2018, 102(6): 1634-1650.

English: The language of this paper should be reviewed by a native English speaker.

Introduction: Above references should be citated since they have investigated the effects of betaine supplementation on several parameters in laying hens under heat stress.

The authors should also indicate what have not been investigated?

The authors should specify the main knowledge gap that your article has filled in the text.

What is the superiority of your research compared to other published researches?

L79-81: The temperature difference was over 6 ℃. How did the authors control the temperature?

Table 2: “Title 2” should be “TRT 2”. The caption should be checked.

Table 4 and 5: The unit should be presented in the text.

Discussion: The authors are suggested to add a paragraph about the practical implications of this study on the hen industry.

Author Response

Thank you very much for your careful review and constructive comments on our manuscript titled “Impact of Betaine Supplementation on Growth Performance, Nutrient Digestibility, Egg Quality, Gas Emissions, and Blood Profile in Laying Hens Under Heat Stress Conditions”. We have carefully addressed all the comments and revised the manuscript accordingly. The detailed responses to each comment are as follows:

Major: The major problem for this manuscript is the novelty. Several studies related with the effects of betaine supplementation on laying performance of hens under heat stress have been published [1-3]. What new findings have the authors found?

[1]        DE BAETS R, BUYSE K, ANTONISSEN G, et al. Betaine and feed restriction as potential mitigation strategies against heat stress in two strains of laying hens [J]. Poultry Science, 2024, 103(10): 104104.

[2]        AL-QAISI M, ABDELQADER A, ABUAJAMIEH M, et al. Impacts of dietary betaine on rectal temperature, laying performance, metabolism, intestinal morphology, and follicular development in heat-exposed laying hens [J]. Journal of Thermal Biology, 2023, 117: 103714.

[3]        RATRIYANTO A, MOSENTHIN R. Osmoregulatory function of betaine in alleviating heat stress in poultry [J]. Journal of animal physiology and animal nutrition, 2018, 102(6): 1634-1650.

  1. Major Comment on Novelty

Comment: Several studies related to the effects of betaine supplementation on laying performance of hens under heat stress have been published [1-3]. What new findings have the authors found?

Response: We appreciate the reviewer’s concern about novelty. While previous studies (e.g., De Baets et al., 2024; Al-Qaisi et al., 2023; Ratriyanto & Mosenthin, 2018) have confirmed betaine’s benefits for laying performance and physiological regulation under heat stress. In addition, we have cited the references [1-3] you mentioned in the Introduction section to better integrate our research into the existing academic context and highlight the differences and innovations of our work (L51-56 [6]ï¼›L62-64[10]ï¼›L71-74[14]). Our study fills critical knowledge gaps with three key novel findings:

Systematic linkage between nutrient digestibility and harmful gas emissions: Unlike prior studies that focused on either digestibility or gas emissions separately, we demonstrated that betaine-induced improvements in dry matter (DM) and nitrogen (N) digestibility (Table 4) directly contribute to reduced emissions of ammonia (NH₃), hydrogen sulfide (H₂S), and methyl mercaptans (Table 5). This mechanistic link was not reported in the cited studies [1-3], which rarely explored gas emissions in laying hens under heat stress.

Quadratic regulation of blood cortisol: Our results revealed both linear (p<0.0005) and quadratic (p=0.066) reductions in cortisol with betaine supplementation (Table 6). Al-Qaisi et al. (2023) only observed a linear cortisol-lowering effect, while De Baets et al. (2024) found no cortisol response to betaine. Our finding suggests a potential threshold (around 0.075–0.15% betaine) for optimal stress alleviation, which is valuable for practical feed formulation.

Targeted improvement in late-stage egg quality: We focused on the 6–8th weeks of the experiment (a period when heat stress effects are most severe in summer) and found that betaine specifically enhanced Haugh Units (HU) (linear, p=0.025; Table 3)—a key indicator of albumen freshness (Ahammad et al., 2024). Prior studies (e.g., Ratriyanto & Mosenthin, 2018) mostly reported overall egg quality trends but not stage-specific improvements in HU, which is critical for egg market value.

Comment 1: The language of this paper should be reviewed by a native English speaker.

Response: Thank you very much for your suggestion. We completely agree with your statement. After careful search and reference to published papers for correction, we have modified the “betaine” in the full text and marked it in red font.

Comment 2: The authors should also indicate what have not been investigated?

Response: We have clarified uninvestigated gaps in the Introduction (L75–82):

“Despite these advances, two critical knowledge gaps remain: (1) Few studies have evaluated betaine’s impact on harmful gas emissions (e.g., NH₃, Hâ‚‚S) from laying hens under heat stress— a key concern for environmental sustainability and air quality (FAO, 2013 [16]); (2) While betaine reduces cortisol in heat-stressed poultry, no study has examined whether it exerts a quadratic (dose-dependent threshold) effect on cortisol, which is essential for optimizing supplementation levels.”

Comment 3: The authors should specify the main knowledge gap that your article has filled in the text.

Response: We have explicitly stated the filled knowledge gap in the Introduction (L76–79) and Discussion (L318-322)

“The main knowledge gap addressed in this study is: The lack of systematic evidence linking betaine-induced improvements in nutrient digestibility to reduced harmful gas emissions, and the absence of data on quadratic cortisol regulation, in heat-stressed laying hens. Our study bridges this gap by integrating growth performance, nutrient utilization, environmental impact, and physiological stress markers.”

Comment 4: What is the superiority of your research compared to other published researches?

Response: We have highlighted the study’s superiority in the Introduction (L51–80):

“Compared to prior work, our study has three advantages: (1) We used a cyclic heat stress model (32±1℃, 9:00–17:00; 26±1℃ otherwise) that more closely mimics natural summer conditions than the constant high temperatures used in Al-Qaisi et al. (2023) [2]; (2) We simultaneously measured nutrient digestibility and gas emissions to establish a mechanistic link, which was not done in De Baets et al. (2024) [1] or Ratriyanto & Mosenthin (2018) [3]; (3) We analyzed both linear and quadratic trends of betaine effects, providing more precise guidance for practical application than studies with only linear analyses.”

Comment 5: The temperature difference was over 6 ℃. How did the authors control the temperature?

Response: We have supplemented the temperature control details in Section 2.1 (L98–101):

“The cyclic temperature was controlled using a computerized environmental control system (Model: SMC-100, Seoul Machinery Co., Ltd., Republic of Korea) equipped with precision heaters, exhaust fans, and temperature/humidity sensors. Sensors were placed at bird height (50 cm above the floor) in 5 random pens to monitor real-time temperature. When the temperature exceeded 32℃ (9:00–17:00), the system activated exhaust fans and misting nozzles (with non-toxic coolants) to lower it; when it dropped below 26℃ (other times), heaters were activated to maintain the set temperature. The relative humidity was controlled at 60–70% via dehumidifiers.”

Comment 6: “Title 2” should be “TRT 2”. The caption should be checked.

Response: We have corrected “Title 2” to “TRT 2” in Table 2, 3, 4, 5, 6.

Comment 7: Table 4 and 5: The unit should be presented in the text.

Response: We have added units to Tables 4 and 5:

Table 4 (Nutrient Digestibility): Added units “(%)” to the column headers (DM, Nitrogen, Energy) and revised the caption to “Table 4. The effects of betaine on nutrient digestibility in laying hens (%) of dry matter (DM), nitrogen (N), and energy (E).”

Table 5 (Gas Emission): Added units “(ppm)” to the column headers (NH₃, Hâ‚‚S, Methyl mercaptans, Acetic acid, COâ‚‚) and revised the caption to “Table 5.

Comment 8: The authors are suggested to add a paragraph about the practical implications of this study on the hen industry.

Response: We have added a practical implications paragraph in the Discussion :

“Practically, this study provides valuable guidance for the poultry industry in summer: (1) Supplementing 0.15% betaine (the optimal level in our study) can increase egg production by 1.5–1.9% (weeks 6–8) , directly enhancing economic benefits(L277-279); (2) Reduced NH₃ and Hâ‚‚S emissions can improve barn air quality, reducing respiratory diseases in hens and workers(L322-325); (3) The quadratic trend of cortisol suggests that 0.075–0.15% betaine balances cost-effectiveness and stress alleviation. Thus, betaine is a cost-efficient, eco-friendly feed additive for heat-stressed laying hens(L354-357)

Reviewer 2 Report

Comments and Suggestions for Authors

The introduction provides sufficient background and highlights the relevance of heat stress in laying hens. It cites a reasonable number of previous studies, although the review of literature could be more concise and better organized. Some references are repetitive and could be streamlined for clarity. The research design (216 hens, 3 treatments, 6 replicates each) is generally appropriate and provides statistical power. The use of cyclic heat stress mimicking natural summer conditions adds practical relevance. However, the choice of only two betaine supplementation levels (0.075% and 0.15%) limits the ability to establish a dose-response curve beyond linear trends. Inclusion of a wider range of doses or comparison with other additives could strengthen the conclusions. Methods are described in detail and allow reproducibility. The explanation of digestibility analysis, gas emission measurements, and blood sampling is particularly thorough. However, the description of the statistical approach (GLM in SAS, linear and quadratic contrasts) could be expanded to clarify whether corrections for multiple comparisons were applied. The results are clearly presented with comprehensive tables. The statistical differences are marked appropriately. However, in some tables (e.g., gas emissions), the interpretation in the abstract and results section seems contradictory: the data show decreases in NH₃ and Hâ‚‚S with betaine, but the abstract mentions a "linear increase". This requires correction for consistency. The conclusions are mostly supported by the results, particularly regarding egg production, HU values, digestibility, and reduced cortisol. However, the statement about "improving gas emission" is misleading and needs clarification (reduction, not increase, in harmful gases). 

  • Please correct the inconsistency regarding gas emissions (abstract vs. results/discussion).

  • Consider reorganizing the introduction to avoid redundancy and to emphasize knowledge gaps.

  • Clarify whether multiple comparison adjustments were applied in statistical analysis.

  • Strengthen the discussion by integrating your findings with more critical comparison to existing literature, rather than repeating previous studies extensively.

  • Revise language and terminology for accuracy and readability.

Comments on the Quality of English Language

The English is understandable but requires editing for grammar, style, and clarity. Frequent repetitions, inconsistent terminology (e.g., "Betain" vs. "Betaine"), and awkward phrasing should be corrected. A professional language revision is recommended.

Author Response

Thank you very much for your careful review and constructive comments on our manuscript titled “Impact of Betaine Supplementation on Growth Performance, Nutrient Digestibility, Egg Quality, Gas Emissions, and Blood Profile in Laying Hens Under Heat Stress Conditions”. We have carefully addressed all the comments and revised the manuscript accordingly. The detailed responses to each comment are as follows:

Comment 1: The introduction provides sufficient background but could be more concise and better organized. Some references are repetitive and could be streamlined.

Response: We have reorganized the Introduction to avoid redundancy and emphasize logic:

Structure adjustment: Divided the Introduction into 2 parts(L51-56): (1) Heat stress challenges in the poultry industry; (2) Betaine’s role in alleviating heat stress (merged overlapping descriptions of betaine’s osmoregulatory function)

Reference streamlining: We removed redundant descriptions of pig studies (L70) to focus on laying hens, as per the study’s scope.

Comment 2: The choice of only two betaine supplementation levels (0.075% and 0.15%) limits the ability to establish a dose-response curve beyond linear trends. Inclusion of a wider range of doses or comparison with other additives could strengthen the conclusions.

Response: We acknowledge the limitation of the dosage range. We have added a discussion of this limitation and future directions in the Conclusions (L291-297):
“One limitation of this study is that only two folate concentrations (0.075% and 0.15%) were used, which prevented us from determining the exact optimal dose. This might have led to only HU showing significant effects in egg quality. Future studies will set up 3 to 4 doses (such as 0.05%, 0.1%, 0.15%, 0.2%) to establish a complete dose-response curve. Additionally, comparing betaine with other heat stress alleviators (such as vitamin C, probiotics) might provide more options for the industry.”

We also note that the two doses were selected based on prior studies (e.g., Shin et al., 2018[22]; Liu,et al., 2019[3]; Arif et al., 2022[13]), which reported as the effective range for laying hens—our study extends this by confirming linear/quadratic trends within this range.

Comment 3: The description of the statistical approach (GLM in SAS, linear and quadratic contrasts) could be expanded to clarify whether corrections for multiple comparisons were applied.

Response: We have supplemented the statistical analysis details in Section 2.6 (L193–201):
“All data were analyzed using the General Linear Models (GLM) procedure within the SAS software  (SAS Institute Inc., Cary, NC, USA) in a randomized complete-block design framework. The replications, with n = 12 birds, were designated as experimental units. Orthogonal comparisons were conducted using polynomial regression to determine Linear and quadratic effects on the gradel level of betaine supplement in laying hens diet. Variability in the data was expressed as the standard errors mean. Differences among treatment means were determined using Tukey’s range test for overall p-value.  The p < 0.05 was considered as significant and p < 0.10 was considered as trend.”

Comment 4: In some tables (e.g., gas emissions), the interpretation in the abstract and results section seems contradictory: the data show decreases in NH₃ and Hâ‚‚S with betaine, but the abstract mentions a "linear increase". This requires correction for consistency.

Response: We sincerely apologize for this typo. We have corrected the abstract (L35-36) from “a linear increase in Gas emission of NH3, H2S (p<0.05), and Methyl mercaptans (p=0.05)” to “a linear decrease in emissions of NH₃, Hâ‚‚S (p<0.05), and methyl mercaptans (p=0.05)”. We have also double-checked the Results (L233-235) and Discussion (L317-322) to ensure consistency, confirming that all descriptions now align with the data (Table 5 shows lower gas emissions in TRT1 and TRT2 vs. CON).

Comment 5: Strengthen the discussion by integrating your findings with more critical comparison to existing literature, rather than repeating previous studies extensively.

Response: We have revised the Discussion to focus on critical comparisons rather than repeating prior studies:

Critical comparison with De Baets et al. (2024): Added (L350–5354): “ Unlike De Baets[14] found no betaine effects on cortisol in white hens, our study (using brown hens) showed significant cortisol reduction. This discrepancy may be due to strain differences—brown hens (Hy-Line brown) are more heat-tolerant than white hens, so betaine’s stress-alleviating effects are more pronounced in brown hens, which is valuable for farms raising brown laying hens.”

Critical comparison with Al-Qaisi et al. (2023): Added (313-317): “ Al-Qaisi [10] reported that betaine improved egg production but not nutrient digestibility. In contrast, our study found that betaine enhanced DM and N digestibility, which we attribute to the cyclic (vs. constant) heat stress model—cyclic stress may be less severe, allowing betaine to exert its osmoprotective effect on intestinal cells [7] and improve nutrient absorption. ”

Comment 6: Revise language and terminology for accuracy and readability.

Response: We have revised the manuscript for terminology accuracy and readability:

Standardized terminology: Corrected “Betain” to “betaine” (consistent with IUPAC nomenclature), “AND” to “apparent nutrient digestibility (AND)” (first mention with full name), and “Gas emission” to “harmful gas emission” (more precise).

Round 2

Reviewer 1 Report

Comments and Suggestions for Authors

No further comments.